# Structural basis for clearing of ribosome collisions by the RQT complex

Katharina Best[1], Ken Ikeuchi ®[1], Lukas Kater[1,2], Daniel Best[1], Joanna Musial[1], Yoshitaka Matsuo ®[3], Otto Berninghausen[1], Thomas Becker[1], Toshifumi Inada[3] ✉ & Roland Beckmann ®[1] ✉

Translation of aberrant messenger RNAs can cause stalling of ribosomes resulting in ribosomal collisions. Collided ribosomes are specifically recognized to initiate stress responses and quality control pathways. Ribosome-associated quality control facilitates the degradation of incomplete translation products and requires dissociation of the stalled ribosomes. A central event is therefore the splitting of collided ribosomes by the ribosome quality control trigger complex, RQT, by an unknown mechanism. Here we show that RQT requires accessible mRNA and the presence of a neighboring ribosome. Cryogenic electron microscopy of RQT-ribosome complexes reveals that RQT engages the 40S subunit of the lead ribosome and can switch between two conformations. We propose that the Ski2-like helicase 1 (Slh1) subunit of RQT applies a pulling force on the mRNA, causing destabilizing conformational changes of the small ribosomal subunit, ultimately resulting in subunit dissociation. Our findings provide conceptual framework for a helicase-driven ribosomal splitting mechanism.

Ribosomal stalling can occur during the translation of messenger RNAs. While transient ribosome stalls may fulfill biological functions, e.g., for co-translational protein folding or targeting, persistent stalls usually indicate a stress situation that requires a response. Such persistent stalling events result in the collision of trailing ribosomes with the stalled ribosome, thereby creating composite interfaces which are then recognized by cells as a proxy for translational stress. Accordingly, these collisions were found to be interaction hubs for numerous factors mediating different translation-based control pathways, such as the integrated stress response (ISR), the ribotoxic stress response (RSR) or the ribosome-associated quality control (RQC) pathway[1–4].

The RQC pathway is triggered after such stalling on rare codons or poly-A stretches, and results in abortion of translation of the aberrant mRNA, endonucleolytic cleavage upstream of the collided ribosomes[5,6], recycling of the stalled ribosome and degradation of the arrested potentially toxic protein product by the ubiquitin-proteasome system[7–10]. Upon ribosome collision, RQC is initiated by

(poly)ubiquitination of several ribosomal proteins near the mRNA entry channel by the E3 ligase Hel2 (ZNF598 in mammals)[11–17]. This is followed by disassembly of the ubiquitinated collided ribosomes by the ribosome quality control triggering complex (RQT) in yeast (ASC-1 or hRQT complex; ASCC in humans)[11,12,18–23] which was shown to first disassemble the lead (stalled) ribosome of the ribosome queue[11]. This event generates a 60S subunit still attached to a peptidyl-tRNA, that is the target of the downstream ribosome-associated quality control (RQC) pathway. Here, in a template-less peptide synthesis event, the RQC-complex adds C-terminal alanine/threonine tails (CAT-tails) via its Rqc2 subunit[24], whereas the E3 ligase Ltn1 polyubiquitinates the nascent peptide for degradation[25–27].

The RQT complex of yeast is composed of the ATP-dependent Ski2-like RNA helicase Slh1 (Rqt2), the ubiquitin-binding CUE domain containing protein Cue3 (Rqt3) and the zinc-finger containing protein Rqt4[11]. The main ribosome splitting activity has been attributed to the helicase Slh1, however, the underlying mechanism of ribosome

[1]Department of Biochemistry, Gene Center, Feodor-Lynen-Str. 25, University of Munich, 81377 Munich, Germany. [2]Friedrich Miescher Institute for Biomedical Research, Maulbeerstrasse 66, 4058 Basel, Switzerland. [3]Division of RNA and gene regulation, Institute of Medical Science, The University of Tokyo, Minato-Ku 108-8639, Japan. ✉e-mail: toshiinada@ims.u-tokyo.ac.jp; beckmann@genzentrum.lmu.de

disassembly by RQT has remained largely enigmatic. The requirement of Hel2-dependent ubiquitination of uS10 and ATP hydrolysis has been shown[11], yet, it is unclear how RQT interacts with collided ribosomes and how it employs its RNA helicase activity for selective splitting of the stalled lead ribosome in a ribosome queue.

Here we show that RQT-driven ribosomal splitting requires binding of RQT to mRNA emerging from the stalled lead ribosome as well as the presence of a neighboring ribosome. Using cryogenic electron microscopy (cryo-EM), we solved structures of several RQT-ribosome complexes, revealing the structural basis of splitting: RQT engages the 40S subunit of the lead ribosome and can switch between two conformations. We propose a mechanistic model in which the Ski2-like helicase 1 (Slh1) subunit of RQT applies a pulling force on the mRNA, causing destabilizing conformational changes of the small ribosomal subunit. The collided ribosome functions as a ram or giant wedge, ultimately resulting in subunit dissociation.

## Results

### mRNA dependent splitting by RQT

We set out to characterize the helicase-dependent splitting activity of RQT biochemically. The closest homolog of its Slh1 helicase subunit is Brr2 (bad response to refrigeration 2), a tandem-helicase cassette containing protein that interacts with single stranded RNA to unwind base-paired spliceosomal U4:U6 snRNA for spliceosome activation[28]. The eponymous Ski2 helicase was shown to bind to mRNA emerging from ribosomes and catalyze the extraction of the mRNA for subsequent mRNA decay by the cytoplasmic exosome[29–31]. Moreover, for the human homolog of Slh1, ASCC3, a direct interaction with the mRNA upon ribosome stalling was recently shown[21]. It is therefore plausible to assume that also the Slh1 helicase of yeast RQT (Fig. 1a) engages mRNA. Taking into account that RQT splits the first/stalled ribosome, we hypothesized that Slh1 requires the presence of an accessible 3′ mRNA overhang emerging from the stalled lead ribosome.

To test if RQT indeed engages mRNA, we optimized our previously described yeast in vitro system[11] to reconstitute and monitor RQT-mediated splitting of ubiquitinated collided ribosomes (Supplementary Fig. 1a) with different mRNA constructs. In brief, a cell-free in vitro translation extract was generated from a yeast strain lacking endonuclease Cue2, exonuclease Xrn1 and RQT component Slh1 ($\Delta cue2$, $\Delta slh1$, $\Delta xrn1$) which has been shown previously to stabilize non-optimal mRNA[5], in order to stably stall ribosomes on an mRNA reporter containing stretches of hard-to-decode CGN (N = A, C or G) codons (Fig. 1b, Supplementary Fig. 1b).

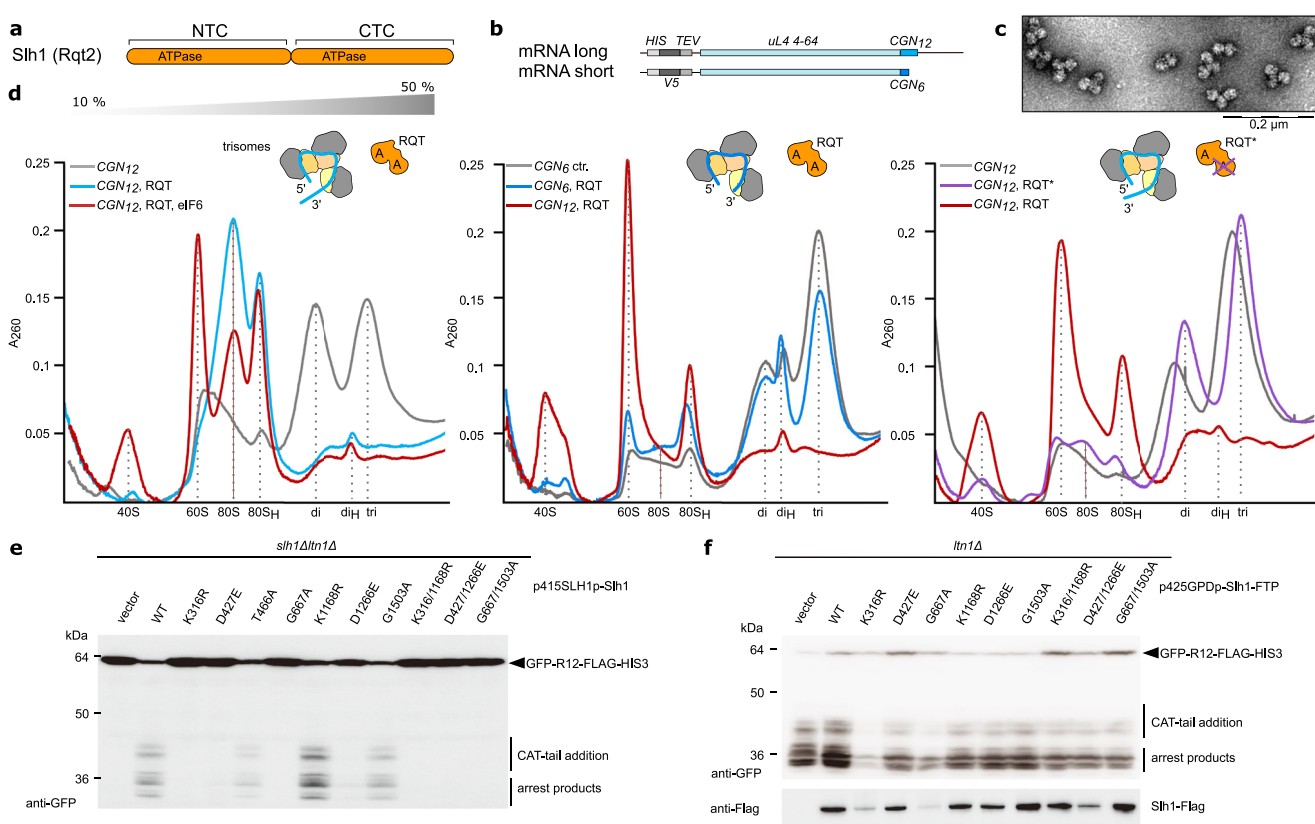

**Fig. 1 | Biochemical analysis of RQT-dependent splitting of collided ribosomes.**
**a** Scheme outlining the tandem ATPase cassettes of Rqt2 (Slh1). NTC, CTC = N-terminal and C-terminal cassette. **b** Scheme outlining the mRNA constructs used for generating ubiquitinated trisomes. $CGN_6$ and $CGN_{12}$ = stalling mRNA sequence containing 6 or 12 consecutive CGN (N = A, G or C) codons. The constructs further encode N-terminal hexahistidine (HIS) and V5 epitope (V5) tags, a Tobacco Etch Virus protease (TEV) cleavage site and a truncated version of ribosomal protein uL4 (uL4). **c** Representative uranyl-acetate stained negative stain-TEM image of poly-ubiquitinated trisomes. We obtained the same results for all (>5) trisome preparations. **d** Polyribosome profiles from sucrose density gradients obtained after in vitro splitting reactions. Left: Splitting reactions using trisomes with a long 3′-mRNA stretch emerging from the mRNA entry site of the lead ribosome (stalled on $CGN_{12}$ mRNA) and of RQT with intact helicase activity (see cartoon representation; A = ATPase). $80_H$ = 80S-40S halfmers; $di_H$ = 40S-disome halfmers. Middle: Comparison of splitting reactions using trisomes with long versus short stretches of 3′-mRNA (stalled on $CGN_6$ mRNA); right: Comparison of splitting reactions using wild type (wt) versus helicase deficient K316R-RQT (RQT*). **e, f** $CGN_{12}$ reporter gene assay to monitor RQC activity in SLH1 mutants: **e** Complementation analysis: SLH1 wild type and the mutants were expressed by endogenous SLH1 promoter in the slh1Δltn1Δ double deletion mutant. **f** Overexpression analysis: SLH1 wild type and the mutants were highly expressed by GPD (TDH3) promoter in the ltn1Δ deletion mutant. The products derived from GFP-CGN(R)12-FLAG-HIS3 reporter gene were monitored by immunoblotting using anti-GFP antibody; Slh1 was probed using anti-FLAG antibody. CAT-tail = C-terminal alanine and threonine tail. For both (**e**) and (**f**), respectively, we obtained essentially the same results in at least three independent experiments. Source data for (**e**) and (**f**) are provided as a Source Data file.

An mRNA reporter was designed such that after stalling at the second or third CGN codon in the P-site[12] an overhang of about 150 nt emerges from the ribosome (($CGN)_{12}$ mRNA) and would be accessible to RQT. After translation of this mRNA construct and subsequent affinity purification of ribosome-nascent chain complexes (RNCs), di- and trisomes were harvested from a sucrose density gradient (Supplementary Fig. 1c), quality controlled by Negative Stain-TEM (Fig. 1c), and in vitro polyubiquitinated using purified Hel2 (E3), Ubc4 (E2), Uba1 (E1), ubiquitin and ATP (Supplementary Fig. 1d, e). Then, splitting reactions with the ubiquitinated ribosome fraction were performed in the presence of ATP and a molar excess of purified wild type RQT (Supplementary Fig. 1f). First, we checked for direct interaction of RQT with the mRNA by employing canonical and 4-thiouridine-containing mRNA for UV-crosslinking during the splitting reaction. Subsequently, Slh1 was affinity purified and associated mRNA analyzed by Northern Blotting. As expected, we observed a substantial enrichment of the mRNA, indicating a direct interaction of the RQT complex with ($CGN)_{12}$ mRNA on collided ribosomes (Supplementary Fig. 1g).

We next checked, whether the 3′-mRNA overhang is necessary for RQT-mediated splitting. We thus designed variants of the *CGN* construct with shorter stall repeats, ($CGN)_6$ and ($CGN)_4$, and no additional 3′ region. Here, only a 9–12 nt (for ($CGN)_6$) or a 3–6 nt long 3′ stretch (for ($CGN)_4$) follows the A-site, and is too short to emerge from the ribosome and could thus not serve as a substrate for RQT. As before, ubiquitinated trisomes carrying these mRNA constructs were incubated with wt RQT or RQT with an ATPase-deficient Slh1-K316R mutant (K316R-RQT; RQT*)[12], as well as purified anti-association factor eIF6. Subsequently, the reactions were analyzed by sucrose density gradient centrifugation. Polyribosome profiles of reactions with ubiquitinated ($CGN)_{12}$ stalled trisomes containing the long accessible mRNA 3′-region show a near complete collapse of the trisome and also disome peak, accompanied by an increase of 40S subunit, 60S subunit and 80S monosome peaks (Fig. 1d, left panel). The control reactions of the same trisomes in the absence of RQT and ATP displayed only a stable trisome peak and an additional disome peak (Fig. 1d), probably as a result of unspecific background nuclease or other dissociation activity. This showed that the RQT complex, as expected, efficiently resolves the collisions by splitting of ribosomes into subunits. Yet, in contrast to the efficiently split ($CGN)_{12}$ trisomes, the splitting activity was largely abolished when using trisomes stalled by the ($CGN)_6$ or the ($CGN)_4$ constructs with a minimal accessible 3′ mRNA overhang (Fig. 1d, middle panel and Supplementary Fig. 1h). This indicated that the RQT activity depends on freely accessible mRNA and supports our hypothesis that the Slh1 helicase activity acts on the mRNA and not the rRNA.

Splitting activity was also lost when using trisomes without prior in vitro ubiquitination (Supplementary Fig. 1i) or when using the ATPase-deficient mutant RQT (Fig. 1d, right panel), consistent with previous studies using *SDD1* mRNA as a reporter[11]. In addition, Walker mutations in both the N- and the C-terminal helicase cassettes (NTCs and CTCs) showed severe effects on the splitting-dependent CAT tailing reaction using a $CGN_{12}$ stalling reporter in vivo. This was observed in complementation assays in which a *SLH1* knockout strain was probed for CAT tailing when complemented with different Slh1 constructs (Fig. 1e). Overexpression of Slh1 constructs in the presence of the wild type gene revealed that two of the NTC mutations display a dominant negative phenotype for CAT tailing (Fig. 1f). Together, this suggested that the activity of both ATPases is required to engage the emerging mRNA and trigger efficient splitting.

We noticed that the second ribosome of a queue was efficiently split in our reaction, most likely following the dissociation of the first ribosome in a processive manner. This was indicated by the almost complete disappearance of not only the trisome but also the disome peak. Importantly, however, the last trailing 80S was not split by RQT since we observed accumulation of 80S ribosomes. This notion is

further supported by the presence of 40S-80S and 40S-80S-80S halfmer peaks, likely representing splitting products in which, after dissociation of the 60S of the lead ribosome, the 40S is still connected with trailing ribosomes via the 40S-40S collision interface or via its mRNA association. In order to corroborate that 80S ribosomes without a collided neighbor ribosome are poor substrates for the RQT complex, we performed the splitting assay with a preparation containing mainly isolated ubiquitinated 80S ribosomes and observed that these individual 80S monosomes were at most modestly dissociated by RQT (Supplementary Fig. 1j). This suggests that the RQT-dependent splitting mechanism requires a neighboring (trailing) collided ribosome which is in line with our observation that the in vitro trisome splitting reactions do not target the last 80S ribosome.

Taken together we conclude that (i) RQT splits with its Slh1 helicase component exerting force primarily on the mRNA, specifically by engaging the 3′ region of the mRNA emerging from the lead ribosome, (ii) that RQT splits in a processive manner and (iii) that RQT requires at least one neighboring ribosome for efficient splitting.

## Cryo-EM of RQT splitting reactions

To obtain deeper mechanistic insights into the splitting process, we performed cryo-EM analysis of our RQT-splitting reactions (Supplementary Fig. 2–6, Supplementary Table 1). To limit the apparent complexity of the trisome sample[11] we used the in vitro ubiquitinated disomes instead of trisomes. We flash-froze the samples quickly after addition of RQT in an attempt to capture reaction intermediates. We expected that reactions containing ubiquitinated disomes and the mutant K316R-RQT or disomes without accessible mRNA overhang (($CGN)_6$-disomes) would enrich pre-splitting intermediates, whereas reactions using the disomes with a long mRNA overhang (($CGN)_{12}$-disomes) and wild type RQT would enrich mainly post-splitting states.

Extensive 3D classification (Supplementary Fig. 2–3) of pre-splitting samples with short mRNA and ATPase-deficient Slh1 showed a large population of intact disomes in their canonical conformation[15,17] prior to splitting: the lead ribosome adopted a non-rotated post-translocational (POST)-state with a P/P tRNA, whereas the trailing ribosome adopted a rotated PRE-state with hybrid A/P-P/E tRNAs. A subpopulation of these disomes was associated with additional density which we could unambiguously assign to the RQT complex with all three subunits (Fig. 2a, b, left). The complex was located on the 40S subunit in close proximity to the mRNA entry tunnel and the ribosomal protein uS10 which has been shown to carry the polyubiquitination modification that signals RQT recruitment[11,12]. While we did not observe density for this modification in any of our structures (probably due to its dynamic nature), the close vicinity between uS10 and RQT is in line with direct interaction licensing RQT for ribosome binding. We found RQT exclusively on the lead but not on the trailing ribosome. This is in agreement with previous biochemical findings[11] and predictions based on kinetic studies[32] suggesting that RQT preferentially splits the lead stalled ribosome of a ribosome queue[11]. This preference can be explained now since in the presence of the lead ribosome it would be impossible for RQT to engage a trailing ribosome without resulting in severe steric clashes; its binding site on the 40 S subunit is almost completely masked by the lead ribosome (Supplementary Fig. 7a). After completed dissociation of the first ribosome, however, RQT could easily engage the next ribosome in a collided queue, explaining the observed sequential activity on the trisome. We noticed that density for the RACK1/Asc1 protein of the lead ribosome found at the collision interface[11,15,17] was underrepresented, probably indicating that destabilizing forces are already in place simply upon ribosome binding of RQT to the collision (Supplementary Fig. 7b, c).

In addition to disomes, these samples contained 80S monosomes, most likely as a result of background dissociation activity; these 80S monosomes adopted the POST-state with RQT bound in the same position on the 40S subunit as in the disome (Fig. 2b). Yet, no small or

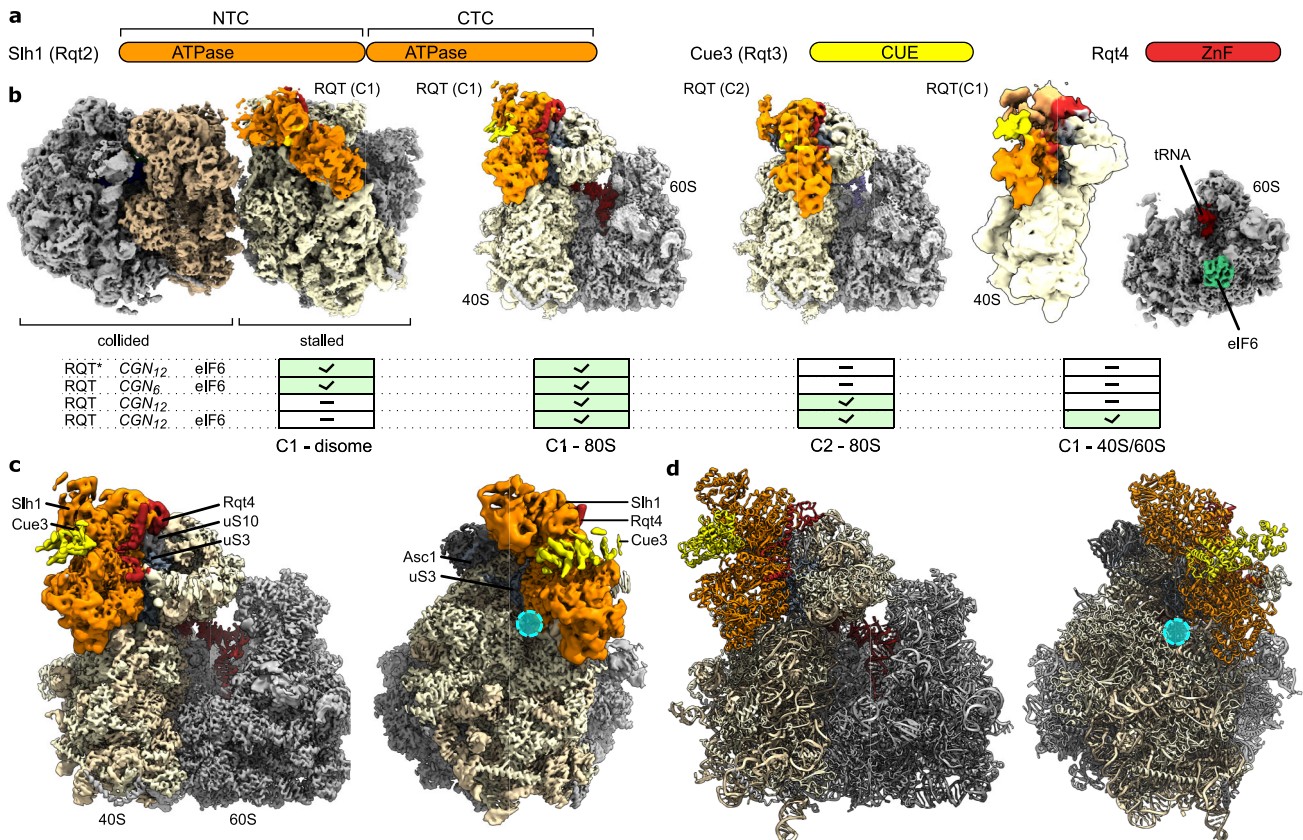

**Fig. 2 | Cryo-EM analysis of RQT-ribosomes complexes. a** Scheme outlining components of the trimeric RQT complex. NTC, CTC = N-terminal and C-terminal cassette, CUE = Coupling of Ubiquitin to ER degradation domain, ZnF = zinc-finger domain. **b** Composite cryo-EM maps of RQT-containing ribosomal particles found in cryo-EM analyses of pre-splitting reactions (short mRNA or ATPase deficient Slh1, RQT*) and post-splitting (long mRNA and wt Slh1) reactions. Occurrence of complexes under different conditions is summarized in the table. C1 and C2 = two different conformations of ribosome-bound RQT. **c, d** Cryo-EM map (**c**) and molecular model (**d**) of the RQT-80S in C1 conformation. All maps are shown as composite maps after local refinement (see Supplementary Fig. 2–5 for details). The mRNA entry site is highlighted with a cyan circle.

large subunits were found when using the ATPase-dead mutants despite the presence of eIF6 in these samples, consistent with the lack of splitting activity observed in the corresponding in vitro assays.

In contrast, when analyzing the samples treated with wt RQT, we mainly observed post-splitting products with almost no stable disomes being detected anymore (Supplementary Fig. 4–5). The majority of particles represented various populations of 80S monosomes and a substantial amount of 40S subunits as well as 60S subunits carrying a peptidyl-tRNA and eIF6 (Fig. 2b), together confirming that splitting reactions took place. Also here, density for RQT was present on 80S monosomes, but contrary to the pre-splitting situation, we found these 80S ribosomes and their associated RQT in two different conformations: the classical POST-state of the lead ribosome (termed C1) (Fig. 2b, c) and a state termed C2 in which the 40S head was swiveled by 20 degrees relative to the 40S body and RQT was repositioned on the 40S (Fig. 2b). This C2 state was present in both eIF6-free but also eIF6-containing datasets indicating that it is not a result of re-association. Moreover, in the presence of eIF6, we also observed a 40S population with RQT bound in the C1 state conformation which may represent the expected splitting product before dissociation of the RQT complex (Fig. 2b). High-resolution refinement of the RQT-bound ribosome classes followed by local refinement of the isolated RQT densities (Supplementary Fig. 6) resulted in a resolution ranging from 3.5 Å in the Slh1 core region to 7.5 Å in the peripheral regions of state C1. This allowed us to build molecular models for RQT-ribosome complexes based initially on AlphaFold 2[33] predictions for the RQT subunits Slh1, Cue3 and Rqt4 (Supplementary Fig. 8–9). These models

reveal the overall architecture of the RQT and how it interacts with the head and body of the 40S subunit in close proximity to the mRNA entry tunnel (Fig. 2d and Fig. 3).

## Structure of ribosome-bound RQT complex

The largest subunit of RQT, the Slh1 helicase, contains, similar to its closest relative Brr2[34], a helical N-terminal domain and a duplicated pair of RecA-like ATPase domains (RecA1 and RecA2), both of which are followed by a winged helix domain (WH) and a Sec63-like domain. The Sec63-like domain is subdivided into Ratchet-, helix-hairpin-helix (HhH) and fibronectin type III (FN3) domains. (Fig. 3a, Supplementary Fig. 8a and Supplementary Fig. 9a).

When bound to the ribosome, Slh1 adopted a bi-lobed arch-like architecture formed by the two RecA-WH-Sec63 cassettes (Fig. 3b, Supplementary Fig. 9b). The N-terminal cassette (NTC) is clamped between the body and the head at the mRNA entry site of the 40S subunit, stretching from helix 16 (h16) of 18S rRNA along the array of ribosomal proteins (r-proteins) eS10, uS3 and uS10 (Fig. 3b, c). The C-terminal cassette (CTC) is packed on the NTC and connects the RQT complex with RACK1 (Asc1) at the back side of the 40S head (Fig. 3b). Densities for both Cue3 and Rqt4 were positioned between the two cassettes with Cue3 binding to the upper convex surface and Rqt4 packing against the concave side of the Slh1 arch close to uS10 (Fig. 3b, e, f and Supplementary Fig. 9b).

Our model for Slh1 accounts for the majority of the 1967 amino acid long protein, lacking only the N-terminal domain (1-217) (Supplementary Fig. 8a). The main anchor of Slh1 in both C1 and C2

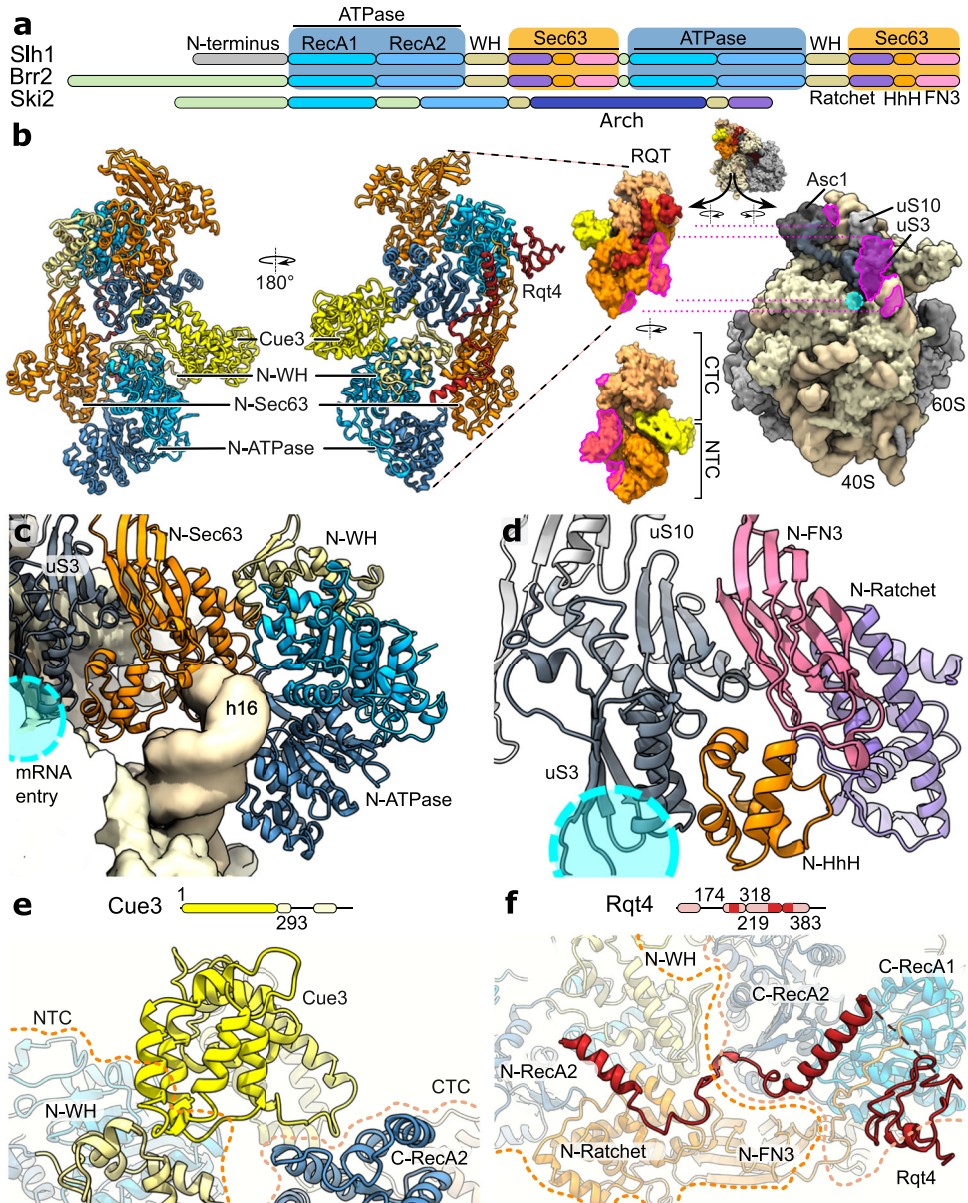

**Fig. 3 | Molecular model of ribosome-bound RQT. a** Domain arrangement of Slh1 and closest homologs Brr2 and Ski2. Slh1 was completely modeled except for the N-terminus (gray). RecA1, RecA2 = RecA-like ATPase domains, WH = winged helix domain, Sec63 = Sec63-like domain (subdivided into Ratchet-, helix-hairpin-helix (HhH) and fibronectin type III (FN3) domains). **b** Molecular model of RQT bound to the 80S ribosome in C1 state. On the right side RQT and the 80S ribosome were converted to low-pass filtered density for clarity. RQT-ribosome interaction regions are highlighted in violet. The mRNA entry site is marked as a blue circle.

A thumbnail indicates the overall orientation NTC, CTC = N-terminal and C-terminal cassette. **c** View focusing on the interaction of Slh1-NTC with the 40S. **d** Zoomed-in view showing the interaction of the N-terminal Sec63 homology region of Slh1 with 40S uS3. **e** View focusing on the Cue3 N-terminal (helical) domain interacting with Slh1. **f** View focusing on Rqt4 interacting with Slh1. In (**e**) and (**f**) schematic representations of Cue3 and Rqt4 are shown. Modeled residues are shown in strong, unmodeled regions in weak color; dashed orange and pale, yellow lines outline NTC and CTC of Slh1.

conformations is constituted by part of the NTC Sec63-domain. Specifically, the HhH domain and the β-sandwich structure of the FN3 domain bind to uS3 at the inner side of the mRNA entry tunnel (Fig. 3c, d). The FN3 domain packs with its terminal β-sheet (which is absent in the CTC FN3 domain) against the β-sheet of the N-terminal uS3 KH domain; the HhH domain is anchored on the two antiparallel α-helices of the uS3 middle domain (Fig. 3d). In C1, the HhH forms additional contacts to eS30, another protein lining the mRNA entry path located in the 40S body near rRNA h16 (Supplementary Fig. 9e). This rRNA helix is a common interaction hub for various ribosome-interacting proteins[35,36] including Ski2[30,31], and is accommodated in a deep pocket formed by the RecA2, Ratchet and HhH domains of the NTC (Fig. 3c). Finally, we observed extra density for the usually flexible uS5

N-terminus located on top of the Slh1 HhH/FN3 domain (Supplementary Fig. 9d).

Compared to the more compact structures of the Brr2 protein (Supplementary Fig. 10f), contacts between the two individual cassettes in Slh1 are limited to a few interactions between the WH domain and a part of the FN3 domain of the NTC and RecA2 of the CTC, forming a patch of mainly hydrophobic interactions (Supplementary Fig. 9c). Thus, Slh1 adopts a rather open, more elongated conformation when compared to Brr2 (Supplementary Fig. 10f). This allows the only direct interaction of the CTC with the 40S, established between its WH domain and RACK1 (Fig. 3b). Moreover, this distinct inter-cassette conformation is stabilized by Cue3 and Rqt4 (Fig. 3b, e, f, Supplementary Fig. 10h): Rod-like extra densities, wedged between the NTC's

WH domain and the CTC RecA2 at the concave side of Slh1, were assigned to the N-terminus (1-297) of Cue3 that forms a helical domain structure as predicted by AlphaFold 2 (Fig. 3e, Supplementary Fig. 8b, Supplementary Fig. 9b, c). This domain of Cue3 is similar to the X-ray structure of the human homolog ASCC2 in complex with the N-terminal domain of ASCC3 (Slh1) which we do not observe[37]. The predicted ubiquitin-binding CUE domain (298-360), a linked four-helix bundle (468-541), and the ultimate C-terminus (542-624) are not visible in our structures, most likely due to their high flexibility. On the opposite convex side of Slh1, and in close vicinity to uS10, density for Rqt4 is visible (Fig. 2, Supplementary Fig. 9b). This density matches well with the predicted C2HC5-type zinc-finger domain (ZFD; 172-218) packing against the CTC RecA1 domain (Fig. 3b, f, Supplementary Fig. 8c, Supplementary Fig. 9b, c). From there, additional, partially helical density extends towards the Slh1 cassette interface bridging the NTC WH domain to the CTC RecA2 domain. Here, the density matched well with the model for a region bridging the C-terminal part of the helical linker with the first helix of the winged-helix-like domain (Supplementary Fig. 8c and Supplementary Fig. 9b). The remaining parts of Rqt4 including the N-terminal PWI-like domain (1-74), two α-helices flanking the ZFD as well as the C-terminus were not visible in our reconstruction due to flexibility.

Taken together, the RQT complex engages the 40S subunit of the ribosome in at least two different conformations, a main conformation C1 and a second conformation C2. In both conformations, RQT adopts an elongated architecture, stabilized by Cue3 and Rqt4 (Supplementary Fig. 10h) with the NTC of Slh1 providing the main anchor point on the ribosome involving mainly uS3 near rRNA h16 in both the C1 and C2 states. Meanwhile, the two accessory subunits Cue3 and Rqt4 bind Slh1 on opposite sides to bridge the NTC and CTC lobes. As a result, the RQT complex is positioned in immediate vicinity to the mRNA entry channel with the RNA substrate paths of its two helicase domains freely accessible to engage mRNA.

## RQT stabilizes unusual 40S conformation

In the C2 state, the 80S ribosome adopted a conformation very similar to a (mRNA/tRNA) translocation intermediate (TI) as described in a recent study analyzing eEF2-bound POST-states[38]. The position of the 40S body relative to the 60S is similar to C1 and thus, closely resembles the classical POST-state. Yet, the 40S head in C2 has undergone a large counterclockwise swivel motion of 20 degrees, resembling essentially the TI-POST-2 state stabilized by eEF2[38] (Fig. 4a–c). While the previously observed translocation intermediate contains two tRNAs in chimeric ap/P and pe/E hybrid states, the RQT-bound C2 state ribosome has only one tRNA in the chimeric pe/E hybrid state as verified by signature interactions of the anticodon-stem loop with U1191 (H.s. U1248) and G904 (H.s. G961) (Fig. 4c).

The RQT in the head-swiveled TI-POST-2 state (C2) showed the same inter-cassette arrangement as in C1, indicating that RQT accompanies the swivel movement (rotation) of the 40S head essentially as a rigid body. It undergoes only a small additional rotation towards the beak of the 40S head by about 10 degrees around its binding site on uS3 (Fig. 4a, b). As a consequence, whereas the main contacts to the 40S head via uS3 and the Slh1 FN3/HhH remain fully intact, the Slh1 NTC relocates on the body of the 40S subunit, with the trajectory of the 40S head swivel corresponding to a downwards movement of the NTC RecA domains of Slh1 along rRNA h16 (Fig. 4b). The RecA2 domain thereby translocates from near the tip of h16 towards its stem. In this position, an additional contact with h16 is formed by the FN3 domain that basically occupies the position of the HhH domain on h16 in the C1 state. In turn, the HhH forms a rRNA contact with h18 at a position that is occupied by the C-terminal tail of eS30 in the C1 state (Supplementary Fig. 9e, f). Taken together, the Slh1 NTC is able to accommodate and stabilize two defined states on the 40S subunit with a distinct head-to-body arrangement.

## Force on mRNA relates to head-swiveling

Our structural observations show that RQT can engage 80S ribosomes in two conformations. Whereas one conformation represented a classical POST-state 80S (C1), the second conformation (C2) represented an unusual late mRNA/tRNA translocation intermediate that so far has only been observed in presence of eEF2 trapped with non-hydrolyzable GMP-PNP and two tRNAs in ap/P and pe/E states[38] (Fig. 4c). During normal translocation, the step leading from the head-swiveled TI-POST-states to the classical POST-state is assumed to be the energy consuming step, where GTP-hydrolysis in eEF2 occurs[38,39]. The transition from our observed RQT-bound states C1 (POST) to C2 (head-swiveled TI-POST-2) thus closely resembles a reverse translocation motion of the 40S head. We speculate that this motion of the 40S head could be key to the RQT-dependent splitting mechanism. Notably, C2 states were neither present in splitting-incompetent reactions including an ATPase-deficient Walker A mutation in the NTC of Slh1(K316R-RQT) nor in reactions with disomes lacking the 3′-mRNA overhang. From this we conclude that an ATP-dependent helicase activity on the mRNA must occur to acquire the C2 state, and that transition to this state is important for splitting of the lead ribosome of the stalled disome.

Interestingly, the transition from the POST-state lead ribosome into a head-swiveled C2 state could be easily achieved by applying force to the mRNA in the opposite direction of mRNA translocation (3′-to 5′-direction)—a pulling mechanism[40]. This notion is also in agreement with the finding that in human cells mRNA damage results in collisions which are resistant to splitting by hRQT[21]. Yet, the question remains whether both or only one of the two connected Slh1 cassettes are responsible for mRNA binding and pulling; based on sequence alignments, contrary to Brr2[41,42], Slh1 may have two active ATPase/helicase sites (Supplementary Fig. 9a). In agreement with this, a contribution of both cassettes was shown for DNA unwinding activity of the human homolog ASCC3[43], although earlier studies assigned this activity to either the N–[37] or the C-terminal cassette[44]. Since we found that single Walker mutations in both the NTC and the CTC abolish RQT activity (Fig. 1e), either both cassettes need to function as helicases, or one of the helicase activities depends on the other cassette's ATPase cycle. Unfortunately, when analyzing the ATPase cassettes for presence of mRNA, no clear density could be identified in either one of them. Yet, when taking into account the relative position of the two cassettes with respect to the mRNA path at the 40S entry site, we speculate that the CTC is more likely to act on emerging mRNA for several reasons: contrary to the CTC and also to Ski2 in the ribosome-bound SKI complex[30,31] the NTC is positioned on the ribosome such that its RNA entry channel is facing away from the 40S mRNA entry site (Supplementary Fig. 10a–d); this becomes clear when comparing ribosome-bound Slh1 to RNA-bound Brr2[45] (Supplementary Fig. 10g). Assuming the same directionality of the helicases and a direct threading of mRNA into the helicase core, the Slh1 NTC activity would result in pushing the mRNA in the 5′–3′ direction rather than applying a pulling force. To realize a pulling (or extraction) activity, the mRNA would need to wind along the inner side of h16 and around the RecA2 domain to enter the helicase core from the other side; this seems sterically highly unlikely. Moreover, in the C1 conformation, direct entry for the mRNA into the NTC helicase core, as observed in the Ski-complex, would be obstructed by rRNA h16 that resides at the interface of RecA1 and RecA2, occupying the space where RNA would exit the helicase (Fig. 4d, Supplementary Fig. 10d, e). In addition, in the C1 conformation, the helicase core of the NTC is in a constricted conformation, mostly resembling the apo/ unengaged states of Brr2 and other related Ski2-like helicases. Therefore, the position and conformation of the Slh1 NTC on the 40S probably prevents engagement with mRNA rather than facilitating it. We speculate that the NTC ATPase activity may be required for the transition on the rRNA h16.

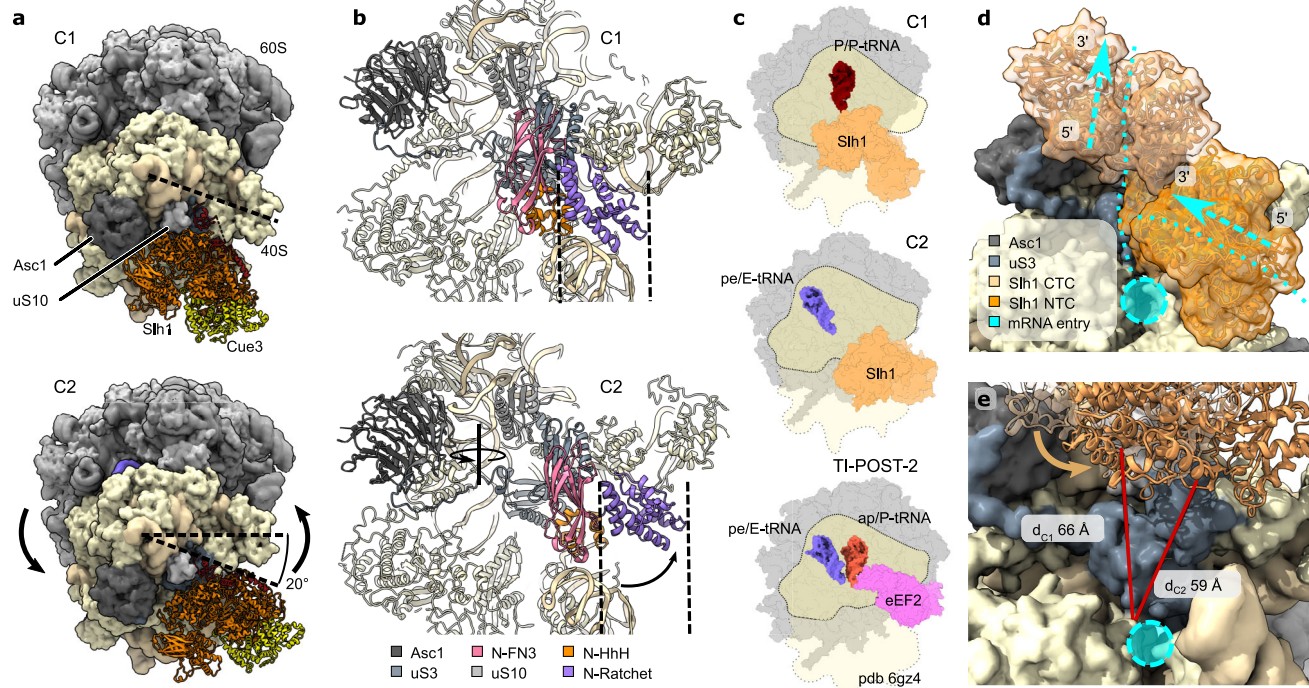

**Fig. 4 | Conformational dynamics of the RQT-ribosome complexes.**
**a**, **b** Comparison of C1 and C2 states of the RQT-80S complex. **a** Top views; in C1, the dashed line marks the position of the 40S head; in C2, the rotation direction and resulting net rotation angle for the head-swivel movement is indicated (20°). For clarity the molecular model of the ribosome was converted to low-pass filtered density. **b** View focusing on the main 40S contacts of the RQT NTC and its repositioning. The solid line in (**b**) displays the rotation axis of the 40S head swivel; RQT undergoes an additional movement and dashed lines assist in assessing the shift of RQT domains relative to the invariant 40S body. HhH = helix-hairpin-helix domain, FN3 = fibronectin type III domain. **c** Top views of cartoon representations

comparing Slh1-bound C1 and C2 80S complexes with the TI-POST-2 translocation intermediate (TI) as observed in the eEF2-GMP-PNP-bound mammalian 80S[38]. **d** Zoomed-in view focusing on the relative position of Slh1 helicase cassettes on the mRNA entry site. Arrows indicate the directions of movement of the 3′–5′-helicase and the mRNA. NTC, CTC = N-terminal and C-terminal cassette. **e** View focusing on the location of Slh1 CTC in C1 (transparent ribbons) and C2 (solid ribbons) states above the mRNA entry channel. Red lines indicate the distance between the helicase entry site and an invariant residue (uS5) of the 40S body nearby the mRNA entry site (cyan circle). Note that Slh1 CTC is 7 Å closer to the mRNA entry site.

We propose that the CTC represents the helicase unit responsible for engaging and applying pulling force on the mRNA. The CTC is positioned ideally to engage with the substrate mRNA in a way such that pulling would occur; and, when superimposing the mRNA path as visible in the ribosome-SKI complex onto RQT-C2, the mRNA would indeed directly thread into the CTC helicase channel (Supplementary Fig. 10e). Moreover, a conformational transition from C1 to C2 corresponds to a shortening of the distance from the CTC helicase unit of Slh1 to the ribosomal mRNA entry by about 7 Å (Fig. 4e). The application of a pulling force to the mRNA by the CTC helicase may be sufficient to trigger the substantial conformational transition from C1 to C2. Importantly, however, we can neither provide direct evidence for the CTC acting as the mRNA helicase nor can we exclude a helicase function of the NTC. The exact roles of the individual ATPase cassettes will require further clarification.

## Discussion

Together with previous findings, our observations allow us to propose an initial working model for the molecular mechanism underlying RQT-dependent splitting of the lead ribosome (Fig. 5).

In the first step, the RQT is recruited to the ribosome queue after Hel2-dependent K63-linked uS10 polyubiquitination via interaction with the CUE domain of the Cue3 subunit[12,22,46,47]. After recruitment, RQT stably locks onto the 40S of the stalled/lead ribosome in the C1 conformation via the Slh1 NTC, placing the Slh1 CTC in a position primed to bind the 3′ region of the mRNA emerging from the ribosomal mRNA entry site. The ATPase-dependent pulling force and (partial) extraction of the mRNA by the Slh1-CTC (and/or NTC) then causes a

head-swivel movement on the lead ribosome and the relocation of Slh1's NTC. This head swiveling results in repositioning the P/P-site tRNA into an ap/P conformation (and the E/E-site tRNA in a pe/E conformation), which likely renders the TI-POST-2 80S substantially less stable than the POST-state 80S; this idea is supported by the observation that this intermediate has only been observed with eEF2 trapped in the presence of non-hydrolysable GMP-PNP. Such a state may lead to a destabilized 80S primed for dissociation of the 40S from the peptidyl-tRNA carrying 60S. Yet, this is not sufficient since we see that for efficient splitting, the presence of a trailing ribosome is necessary.

According to our model, head-swiveling may directly exert force on the trailing collided ribosome. Along the trajectory of the swiveling motion, RACK1/Asc1 of the lead ribosome would clash with RACK1/Asc1 of the trailing ribosome. Notably, we observed that prolonged binding of RQT to collided ribosomes with truncated mRNA is already sufficient to destabilize this RACK1/Asc1-RACK1/Asc1 contact (Supplementary Fig. 7d, e; Supplementary Movie 1). In case of intact disomes, head-swiveling of the lead ribosome would force the trailing ribosome to accommodate this movement by rotating in the opposite direction with the connecting mRNA between the two ribosomes serving as a pivot. This results in a net movement of the trailing ribosome in the direction of the 60S subunit of the lead ribosome, with a primary impact on the contact of the 40S-body to the stalled 60S of the disome. In this situation, we speculate that the 60S of the lead ribosome will be dislocated from its position, leading to further destabilization of the lead ribosome and eventual dissociation of the 60S subunit from the 40S. Moreover, in case of prolonged RQT helicase activity and extended pulling on the mRNA, the trailing ribosome would ultimately

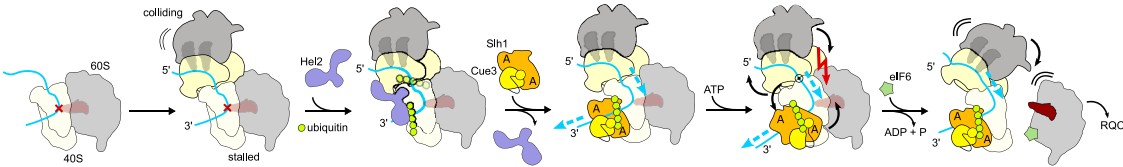

**Fig. 5 | Model for RQT-mediated ribosome splitting after collision.** After stalling and subsequent collision of trailing ribosomes, Hel2 recognizes the collision interface and (poly)ubiquitinates uS10. RQT binds polyubiquitinated stalled ribosomes and engages with accessible 3′-mRNA emerging from the stalled lead ribosome. A pulling force on the mRNA (blue arrows) may cause head-swiveling, destabilization of the lead ribosome and ultimately drive the trailing ribosome like a wedge between the subunits of the lead ribosome. The red cross indicates a stall on problematic mRNA (blue); P-site tRNA in the stalled ribosome tRNA is indicated in transparent red and hybrid tRNAs in the collided ribosomes are transparent gray; Green circles indicate ubiquitin; black arrows indicate possible directions of movement and black curved lines indicate movement/flexibility; A = ATP-binding site. See text for details.

serve as a giant wedge that is driven between the subunits of the lead ribosome. Yet, since the mRNA connecting the collided ribosomes must be already under tension, as indicated by the incomplete tRNA translocation of the trailing ribosomes, we predict that any additional force application, extending the mRNA by only a few nucleotides, should have an immediate destabilizing effect.

The proposed RQT mechanism is reminiscent of the remodeling of the spliceosome by the two related RNA helicases Brr2 and Prp22. Brr2 engages and translocates a single-stranded region of U4 snRNA[48], and Prp22 applies pulling force on single stranded mRNA for its release[40]. However, a ribosome splitting mechanism that is driven by an mRNA helicase and that works through destabilizing conformational transitions of the ribosome, represents a different principle. For ribosome recycling after canonical translation termination, bacteria employ the GTPase EF-G together with the cofactor RRF (ribosome recycling factor), which functions by driving the cofactor as a splitting wedge between the ribosomal subunits[49–51]. The ABC-type ATPase ABCE1 works in a similar way in *Archaea* and eukaryotes, also by causing steric clashes between the subunits and cofactors aRF1, eRF1 and Dom34, respectively[52,53]. However, the ABCE1 machinery cannot function on stalled collided ribosomes as long as a tRNA in the A-site or a translation factor prevents cofactor binding. This may explain the necessity for the evolution of an alternative splitting mechanism employing a mechanism that can function on any stalled ribosome irrespective of its A-site occupancy. Importantly, the observed mechanism is highly specific in that it works only on ribosomes with a collided neighbor and requires licensing/recruitment through ubiquitin tagging of the stalled ribosome. These ideas are in line with the recent discovery in the bacterium *Bacillus subtilis* of an alternative splitting factor, the ATPase MutS2, that clears collided ribosomes[54]. Although its mode of function is not entirely clear yet, it appears to function independent of mRNA, thereby making the RQT mechanism a unique example of an mRNA helicase-dependent ribosomal splitting device which is conserved in eukaryotes from fungus to humans. Further studies on RQT will shed light on the exact mode of mRNA engagement and on the role of the ATP hydrolysis cycle of the individual helicase units in promoting ribosome splitting.

## Methods

### RQT purification

A yeast strain overexpressing RQT components (Slh1-FTP, Cue3, Rqt4, see Supplementary Table 2) was grown in synthetic drop-out media. Cells were harvested by centrifugation and subsequently disrupted by cryogenic milling. Cell powder was resuspended in lysis buffer (50 mM $K_2HPO_4$ /$KH_2PO_4$ pH 7.5, 500 mM NaCl, 5 mM Mg(OAc)$_2$, 100 mM arginine, 1 mM DTT, 1 mM PMSF, 0.1% NP-40, 1 protease inhibitor pill/ 50 ml (Roche, #04693132001)) and centrifuged at 30,596 x $g$ for 30 min. Purification was performed on IgG-sepharose (GE Healthcare, #GE17096901) resin. The complex was eluted by TEV cleavage in RQT elution buffer (50 mM HEPES pH 7.5, 300 mM KOAc, 5 mM Mg(OAc)$_2$, 0.01% Nikkol, 1 mM DTT) for 1.5 h at 4 °C.

### eIF6 purification

*S. cerevisiae* eIF6A was purified essentially as described before[55] from the p7XC3GH (Addgene #47066; Watertown, MA, USA) plasmid, expressing eIF6 with a C-terminal 3 C protease cleavage site, GFP, and His$_{10}$-tag from *E. coli* Rosetta cells. The plasmid was transformed into *E. coli* Rosetta (DE3), which was grown at 37 °C to mid-log phase. The temperature was reduced to 16 °C, and protein overexpression was induced with 100 uM IPTG for overnight expression. Cells were harvested at an OD$_{600}$ of 4 (4000 × g, 4 °C, 10 min), washed with phosphate-buffered saline (PBS), and resuspended in ca. 100 ml lysis buffer (20 mM Tris-HCl pH 8.0, 300 mM NaCl, 2 mM β-ME, 1x cOmplete EDTA-free Protease Inhibitor Cocktail tablets (Roche, #04693132001)) before being lysed with Microfluidics M-110L microfluidizer. Lysates were clarified by centrifugation at 30,596 × g at 4 °C for 30 min. Clarified lysate from 2 l cell culture, (ca. 12 g cells) was loaded onto 2.5 ml (5 ml slurry) TALON metal affinity resin (Takara Bio) equilibrated in lysis buffer and incubated on a rotating wheel at 4 °C for 1 h. The supernatant was removed, and the resin was washed three times with lysis buffer containing 10 mM imidazole, before being incubated with ca. 6 ml elution buffer (20 mM Tris-HCl pH 8.0, 300 mM NaCl, 2 mM β-ME, 10 mM imidazole, 0.25 mg/mL 3 C protease) for 30 min at 4 °C. Eluted protein was concentrated to 1 mL before being loaded onto Superdex 200 (Sigma-Aldrich) for size exclusion chromatography in the final buffer (50 mM HEPES pH 7.5, 500 mM KCl, 2 mM MgCl$_2$, 2 mM β-ME).

### Ubc4 purification

Recombinant yeast Ubc4 was purified essentially as described before[17] as GST-3C-Ubc4 from *E. coli* Rosetta2(DE3) harboring pGEX-UBC4 and was eluted as non-tagged Ubc4 by PreScission Protease (Cytiva Cat# GE27084301). *E. coli* cells were cultivated in 2 l of 2xYT medium with 100 μg/ml Ampicillin and 100 μg/ml Chloramphenicol at 37 °C until 0.7 of OD$_{600}$. IPTG was added to the culture at final concentration of 0.1 mM and further incubated at 20 °C for 20 h. Cells were harvested by centrifugation and the cell pellet was frozen by liquid nitrogen. Cells were ground by mortar and pestle in liquid nitrogen. Resulting cell powder was lysed in lysis buffer (50 mM Tris-HCl pH 7.5, 300 mM NaCl, 5 mM MgCl$_2$, 1 mM DTT, 0.5 mM PMSF, 0.01% NP-40, 10 unit/ml of DNase-I (Invitrogen Cat# 18047019)) and the lysate was cleared by centrifugation at 40,000 x $g$, 40 °C for 15 min in SS-34 rotor. Lysate of *E. coli* cells harboring pGEX-UBC4 was incubated with Glutathione Sepharose 4B (GE Healthcare Cat# 17-756-05) followed by wash with lysis buffer without MgCl$_2$ and DNase-I for four times, and elution buffer (50 mM Tris-HCl pH 7.5, 100 mM NaCl, 5 mM MgCl$_2$, 1 mM DTT) for three times. Ubc4 was eluted from beads using 80 μl Pre-ScissionProtease in 1 ml elution buffer at 4 °C for 16 h.

### Hel2 purification

A yeast strain (see Supplementary Table 2) overexpressing Hel2-FLAG was grown in synthetic dropout media. Cells were harvested by centrifugation and subsequently disrupted by cryogenic milling (SPEX

SamplePrep 6970EFM Freezer/Mill). Cell-powder was resuspended in lysis buffer (50 mM Tris pH 7.5, 500 mM NaCl, 10 mM Mg(OAc)$_2$, 0.01% NP-40, 1 mM PMSF/DTT, 1 protease inhibitor pill/50 ml (Roche, #04693132001)) and centrifuged at 30,596 x $g$ for 30 min. The supernatant was purified using M2 FLAG affinity resin (Sigma-Aldrich, #A2220). During the washing steps, the salt concentration was decreased from 500 mM NaCl to 100 mM NaCl in 100 mM steps. Hel2-FLAG was eluted by incubation of the resin with 3x FLAG peptide in elution buffer (50 mM HEPES, 100 mM KOAc, 5 mM Mg(OAc)$_2$, 1 mM DTT, 0.05% Nikkol).

## In vitro translation of stalling constructs

$CGN_{12}$- $CGN_6$- and $CGN_4$-reporters were generated by PCR using the plasmid pEX-HIS-V5-uL4A-CGN12 as template. It expresses a truncated version of ribosomal protein uL4[56] followed by a mRNA-based stalling sequence containing twelve consecutive arginine-encoding CGN (N = A, G or C) codons ($CGN_{12}$; "R12 cassette") and a 3'-region of 129 bases. $CGN_6$ and $CGN_4$ constructs were derived by truncation after the sixth or fourth CGN codon. In vitro translation of these constructs would result in accumulation ribosomes stalled after the second or third CGN codon in the P-site[12] with long ($CGN_{12}$) or short ($CGN_6$; $CGN_{12}$) 3'-mRNA overhangs emerging from the mRNA entry channel of the stalled ribosome (see Fig. 1 and Supplementary Fig. 1).

*HIS-V5-uL4-CGN* mRNAs were obtained using a mMESSAGE mMACHINE T7 kit (Invitrogen #AM1344) using linear DNA templates. Cell-free yeast translation extracts were prepared as described previously using Δ*xrn1*Δ*slh1*Δ*cue2* or Δ*ski2*/*uS10-HA* yeast strain[57,58]. In brief, strains were freshly steaked on YP plates and incubated at 30 °C for 2 days. From the plate, pre-cultures were incubated at 30 °C. The start OD$_{600}$ was chosen such that the cells are at an OD$_{600}$ of ~2 at the time of harvest. The cells were grown in 10–20 l YP medium with 2% glucose, supplemented with both ampicillin and anti-foam. Cultures were harvested by centrifugation, resuspended in ice cold water, pooled, spun down to remove the water and pelleted. The yeast cells were either frozen as droplets in liquid nitrogen and cell walls were removed by cryogenic milling and subsequent centrifugation steps, or cell free extract was prepared directly via preparation of spheroblasts. The in vitro translation reaction was performed in the presence of Hel2 (50 nM) at 17 °C for 75 min. Ribosome nascent chain complexes (RNCs) were purified via the encoded His-Tag using magnetic Dynabeads (Invitrogen, #10104D). The translation reaction was applied to equilibrated beads (50 mM HEPES pH 7.5, 300 mM KOAc, 10 mM Mg(OAc)$_2$, 125 mM sucrose, 0.01% NP-40, 5 mM β-mercaptoethanol) for 20 min, washed and eluted in elution buffer (50 mM HEPES pH 7.5, 150 mM KOAc, 10 mM Mg(OAc)$_2$, 125 mM sucrose, 0.01% NP-40, 5 mM β-mercaptoethanol) containing 400 mM imidazole.

## Isolation of di- and trisomes

Purified RNCs were loaded on top of 10–50% sucrose gradients (50 mM HEPES pH 7.5, 200 mM KOAc, 10 mM (MgOAc)$_2$, 1 mM DTT, 10/50% sucrose (w/v)) followed by ultracentrifugation for 3 h at 284,000 x $g$. Fractions of the gradients were collected on a gradient station (Biocomp) equipped with a TRIAX flow cell (Biocomp) and a GILSON fractionator. Mono-, di- and trisome fractions were used for further experiments.

## In vitro ubiquitination of stalled RNCs

The in vitro ubiquitination reaction was performed essentially as described before with adjustments[17]. 10-15 pmol of purified RNCs, or isolated di-/trisomes were incubated with 57 μM ubiquitin, 116 nM UBE1 (R&D systems), 3.3 μM Ubc4 and 757 nM Hel2 in a reaction buffer (20 mM HEPES-KOH pH 7.4, 100 mM KOAc, 5 mM (MgOAc)$_2$, 1 mM DTT, 1 mM ATP, 10 mM creatine phosphate, 20 ug/ml creatine kinase (Roche)) at 25 °C for 30 min. Ubiquitinated ribosomes were either used directly in in vitro splitting assays or pelleted through a sucrose

cushion (50 mM HEPES pH 7.5, 100 mM KOAc, 25 mM Mg(OAc)$_2$, 1 M sucrose, 0.1% Nikkol) in a TLA110 rotor (Beckman Coulter) at 434,513 x $g$ for 1.5 h at 4 °C, for the subsequent preparation of cryo-EM samples.

## Semi-dry Western blotting

After sample separation on NuPAGE, semi dry western-blotting was performed. The PVDF membrane was blocked with 5% skim milk in TBS for 1 h and incubated with anti-HA-HRP (Roche, 3F10, 1:5000) in 5% milk/TBS. After washing (1x TBS with 0.1% (w/v) TWEEN-20, 2x TBS) signal was detected with an AI-600 imager (GE Healthcare) using SuperSignal West Dura Extended Duration Substrate (Thermo).

## In vitro splitting assays

6-12 pmol of RNCs (ubiquitinated or non-ubiquitinated) were incubated with 10x molar excess of RQT complex (wild type or K316R-Slh1 mutant; RQT*) and 5x molar excess of eIF6 for 15 min at 25 °C. For control reactions without RQT, RQT elution buffer (50 mM HEPES pH 7.5, 300 mM KOAc, 5 mM Mg(OAc)$_2$, 0.01% Nikkol, 1 mM DTT) was added instead. The reactions were separated on a 10–50% sucrose gradient using ultracentrifugation in a SW40 rotor for 3 h at 284,000 x $g$.

## mRNA interaction assay

For analysis of the RQT-mRNA interaction, 4-thiouridine (4SU)-containing mRNA was generated by addition of 7.5 mM 4SU to the in vitro transcription reaction. 8 pmol of ubiquitinated RNCs (stalled on $CGN_{12}$ or $CGN_{12}$-4SU mRNA) were incubated with a 10x molar excess of purified RQT complex at room temperature for 5 min. After crosslinking with UVA, samples were affinity-purified using M2 FLAG affinity resin (Sigma-Aldrich, #A2220). Binding of Slh1-ribosome complexes to the beads was performed in binding buffer (50 mM HEPES pH 7.5, 200 mM KOAc, 100 mM arginine, 125 mM sucrose, 1 mM PMSF) at 4 °C for 1 h. After two washing steps with binding buffer, again two washing steps with either 4 mM ATP or 5 mM EDTA were performed. Samples were eluted by addition 3xFLAG peptide in elution buffer (50 mM HEPES, 100 mM KOAc, 62.5 mm sucrose, 1 mM DTT). To remove remaining protein, samples were treated with proteinase K in ProtK buffer (100 mm Tris-HCl pH 7.5, 50 mm NaCl, 10 mM EDTA pH 8) at 37 °C for 20 min at a concentration of 1 mg/ml. Subsequently, equal volume of ProtK buffer containing 7 M urea was added for further incubation at 37 °C for 20 min. After addition of TRI reagent (Zymo Research #R2051), RNA was extracted using a Direct-zol Mini Prep kit (Zymo Research #R2051).

## Northern Blotting

6 μL of purified RNA was mixed with 19 μL of RNA loading buffer (30 mM Tricine, 30 mM Triethanolamine, 0.5 M formaldehyde, 5%v/v glycerol, 1 mM EDTA, 0.005%w/v Xylene cyanole, 0.005%w/v bromophenol blue in deionized formamide) and incubated at 65 °C for 5 min followed by resting on ice for 5 min. RNA was separated on a 1.2% agarose gel with 1x TT buffer (30 mM Tricine, 30 mM Triethanolamine, pH 7.9 in 50X stock) by electrophoresis at 120 V for 100 min, followed by capillary transfer onto a Hybond-N + membrane (cytiva #RPN303B) using 20X SSC (3 M NaCl, 300 mM Sodium citrate). Hybridization was performed at 52 °C using DIG-labeled $R(CGN)_{12}$ probe (5'-DIG-GCGGCGCCGTCGTCGCCGGCGGCGCCGTCGTCGCCGTTCCCAGGATT CAG-3') and DIG easy hyb granules (Roche # 11796895001) and incubated in hybridization oven for 20 h. The membrane was washed once for 15 min by 2X SSC 0.1% SDS, twice for 15 min each by 0.1X SSC 0.1% SDS in the oven, then incubated in blocking reagent (Roche #11096176001) for 30 min and with 1/10,000 anti-digoxigenin-AP (Roche #11093274910) for 1 h at room temperature. After washing membrane three times by wash buffer (100 mM maleic acid, 150 mM NaCl, 0.3% Tween-20, pH 7.5) and once by pre-detection buffer (0.1 M Tris-HCl pH 9.5, 0.1 M NaCl), RNA was detected via chemiluminescence

using CDP star reagent (Roche #11759051001) on an AI-600 mini (GE healthcare) for 24 min.

### $CGN_{12}$-reporter gene assay to probe for CAT-tailing

Exponentially grown yeast cultures were harvested at $OD_{600}$ of 0.5‑0.8. Cell pellets were resuspended with ice-cold TCA buffer (20 mM Tris pH8.0, 50 mM $NH_4OAc$, 2 mM EDTA, 1 mM PMSF, 10% TCA) and then an equal volume of 0.5 mm dia. zirconia/silica beads (BioSpec) was added followed by thorough vortexing for 30 s, three times at 4 °C. The supernatant was collected in a fresh tube. After centrifugation at 18,000 x $g$ at 4 °C for 10 min and removing supernatant completely, the pellet was resuspended in TCA sample buffer (120 mM Tris, 3.5% SDS, 14% glycerol, 8 mM EDTA, 120 mM DTT and 0.01% BPB). Proteins were separated by SDS-PAGE and were transferred to PVDF membranes (Millipore; IPVH00010). After blocking with 5% skim milk, the blots were incubated with the anti-GFP antibody (Santa Cruz Biotechnology, Cat# sc-9996, clone B-2; dilution 1:5000), followed by the $2^{nd}$ incubation with the anti-mouse IgG antibodies conjugated with horseradish peroxidase (Cytiva, Cat# NA931 ECL; dilution 1:5000). The products derived from $CGN_{12}$ reporter gene were detected by homemade ECL solution using the ImageQuant LAS4000 (GE Healthcare). Overexpressed Slh1 was probed using anti-FLAG antibody (Sigma-Aldrich, Cat# F1804, clone M2; dilution 1:5000)

### Cryo-EM samples of RQT-ribosome complexes

To generate suitable samples for cryo-EM, at least 6 pmol of ubiquitinated disomes were incubated for 5 min at 25 °C with 12 pmol of purified RQT complex in the presence of 1 mM ATP. After incubation, 3.5 μl of samples were vitrified in liquid ethane on glow discharged, R3/3 copper grids with a 2 nm carbon coating (Quantifoil) using a Vitrobot mark IV (FEI) with 45 s wait time and 2.5 s blotting time.

Altogether four samples were analyzed, two pre-splitting reactions and two post-splitting reactions. Pre-splitting reactions contained either ubiquitinated $CGN_6$-stalled disomes (with short accessible 3′-mRNA) and wild type RQT ($RQT_{wt}$) or (ubiquitinated) $CGN_{12}$-stalled disomes (with long accessible 3′-mRNA) and the RQT mutant containing K316R-Slh1 (RQT*) as well as eIF6. Post-splitting reactions contained ubiquitinated $CGN_{12}$-stalled disomes and wild type RQT, one reaction with and one without eIF6.

For the post-splitting sample (with $CGN_{12}$-stalled disomes and $RQT_{wt}$ without eIF6 addition) sample, 21.171 movies were collected on a Titan Krios with a K2 Summit DED, at 300 keV with a pixel size of 1.045 Å/pixel. The applied electron dose was ‑1.09 e⁻/Å/frame for 40 frames and data were collected in a defocus range between 0.5–3.5 μm. All frames were gain corrected, aligned and subsequently summed using MotionCor2[59,60]. Downstream data processing was performed using CryoSPARC (v.3.3.1)[61,62]. The CTF was estimated using gCTF and CTFFIND4[63,64], followed by particle picking via CryoSPARCs blob picker. After 2D classification, 2.415.630 particles were used for 3D refinement. Subsequent rounds of classification were carried out using 3D Variability Analysis[61] with a soft mask around the ribosomal 40S subunit. Subsequent sorting steps with a soft mask around the RQT complex yielded two classes containing RQT in two distinct conformations (C1 and C2). The class of 80S ribosomes with RQT bound in C1 contained 194.186 particles (8% of total particles) and was refined to a overall resolution of 2.4 Å. Subtracted particles of RQT were locally refined to a resolution of 3.5 Å. The second class of 80S with RQT bound in C2 contained 20.380 particles and was refined to a overall resolution 3.0 Å. Local refinement was carried out as for C1 and resulted in a resolution of 4.8 Å. The processing scheme and (local) resolution estimations for sample are depicted in Supplementary Fig. 4 and Supplementary Fig. 6, respectively.

The other three samples were vitrified and cryo-EM data were obtained as described above. For the $CGN_{12}$/RQT* pre-splitting sample, 11.251 movies were collected with an applied electron

dose of ‑1.09 e⁻/Å/frame for 40 frames. For $CGN_6$/$RQT_{wt}$/eIF6 presplitting sample, 16.508 movies were collected with an applied electron dose of ‑1.1 e⁻/Å/frame for 40 frames. For $CGN_{12}$/$RQT_{wt}$/eIF6 post-splitting sample, 14.092 movies were collected with an applied electron dose of ‑1.16 e⁻/Å/frame for 40 frames. Downstream processing was performed using CryoSPARC (v3.3.1) as depicted in Supplementary Figs. 2–5.

### Model building and refinement

The RQT-ribosome C1 model was prepared by rigid body docking the model for $SDD1$ stalled 80S (PDB code 6SNT[11]) and the models predicted by AlphaFold 2[33,65] for RQT components Slh1 (Uniprot-ID P53327), Cue3 (Uniprot-ID P53137) and Rqt4 (Uniprot-ID P36119). For Slh1, the N-terminal and C-terminal cassettes (NTC; residues 217-1122 and CTC residues 1123-1967) were docked individually. For Cue3 only the N-terminal part (residues 1-297) and for Rqt4, two parts (residues 171-219 and 323-381) were docked (see also Supplementary Fig. 8).

To obtain the RQT-ribosome C2 model, 60S, 40S and RQT of the RQT-ribosome C1 model were individually rigid-body docked into the cryo-EM density. For the collided ribosomes, 40S and 60S of the $SDD1$ stalled 80S model (PDB code 6SNT) were docked individually. tRNAs were taken from the model of the yeast disome (PDB code 6I7O)[17] and rigid body docked into the A/P-, P/E-site densities.

The 60S model was prepared by docking the model for NatA-bound 60S (PDB code 6HD7)[56] into the cryo-EM density. tRNA was taken from 6SNT, eIF6 from PDB code 1G62[66]

Adjustment of all models was performed using Wincoot (v.0.9.6) and subsequently real-space refined using Phenix (1.19.1)[67] To obtain figures, molecular models and cryo-EM densities were displayed in ChimeraX (v.1.3)[68,69].

### Reporting summary

Further information on research design is available in the Nature Portfolio Reporting Summary linked to this article.

## Data availability

The cryo-EM structural data generated in this study have been deposited in the Protein Data Bank and the Electron Microscopy Data Bank under accession codes EMD-14861 and PDB-7ZPQ for the RQT-80S in C1 conformation; EMD-15280 for isolated, local refined RQT in C1 conformation, EMD-15228 for 80S in C1 conformation; EMD-14921 and PDB-7ZRS for the RQT-80S in C2 conformation; EMD-14926 and PDB-7ZS5 for the 60S-peptidyl-tRNA complex; EMD-14978 and PDB-7ZUW for the lead ribosome in the RQT-disome complex in C1 conformation; EMD-14979 and PDB-7ZUX for the collided RQT-80S ribosome in the RQT-disome complex. Source data are provided with this paper.

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

## Acknowledgements

We thank Charlotte Ungewickell, Susanne Rieder and Alicia Musial for excellent technical assistance, Dr. Rachel Green for kindly providing the Δ*cue2*, Δ*slh1*, Δ*xrn1* yeast knockout strain. Dr. Petr Tesina and Timo Denk for discussions and Dr. Jingdong Cheng for advice in model building.

This work was supported by grants by DFG to R.B. (BE 1814/15-1, RTG1721), a DFG fellowship through the Graduate School of Quantitative Bioscience Munich (QBM) to K.B., the European Research Council grant ADG under the European Union's Horizon 2020 research and innovation program (Grant agreement No. 885711— Human-Ribogenesis) to R.B., JSPS Overseas Research Fellowship to K.I., grants byAMED (grant JP19gm1110010 to T.I.) and MEXT/JSPS KAKENHI under Grant Numbers JP19H05281, 21H05277 and 22H00401 to T.I., 21H00267 and 21H05710 to Y.M., by Takeda Science Foundation to T.I and by JST PREST Grant Number JPMJPR21EE to Y.M.

## Author contributions

R.B., T.I., K.B., and T.B. designed the study; O.B. collected cryo-EM data; K.B. and L.K. prepared cryo-EM samples and processed cryo-EM data; K.B. built molecular models; Y.M. performed the (*CGN*)$_{12}$ reporter gene assay. K.B., L.K. and D.B. performed in vitro translation and splitting assays with help from J.M. and Y.M.; K.B. and K.I. performed in vitro ubiquitination and Northern Blots. K.I. and Y.M. generated plasmids and yeast strains. K.I., K.B. D.B. and J.M. performed protein purifications; T.B., R.B. T.I. and K.B. wrote the manuscript, with comments from all authors.

## Funding

## Competing interests

The authors declare no competing interests.
