## [Peer Review File · Nature Communications]

Structural basis for clearing of ribosome collisions by the RQT complexREVIEWER COMMENTS

Reviewer #1 (Remarks to the Author):

Ribosome collisions are recognized by the ribosome-associated quality control system and important for rescuing stalled ribosomes in all domains of life. Dissociation of the collided ribosomes is mediated by the ribosome quality control trigger complex RQT although the mechanism by which this is mediated is unknown. Here the authors provide insight into this process using a combination of in vitro biochemical splitting assays and determination of cryo-EM structures of several RQT-ribosome complexes. RQT is shown to require accessible 3' extension of the mRNA and the presence of a neighboring ribosome. Cryo-EM structures reveal that RQT engages the 40S subunit of the lead ribosome and provide the first overall architecture of RQT on the ribosome. Moreover, two conformations for RQT are observed that lead to a model whereby the RQT component Slh1 applies a pulling force on the mRNA that imparts a destabilizing conformation on the 40S subunit, with contribution of the collided ribosome that acts as a wedge to aid dissociation.

Overall, this an exciting study with providing not only a first insights into the overall architecture of RQT on the ribosome, but also mechanistic insights into how RQT mediates splitting. The manuscript is well-written and presented and the results will be of great interest to the scientific community.

I have no major concerns but only a few points for clarification.

1. In the main text, the authors skirt around the resolution of the complexes and in particular the local resolution of the factors, therefore, the reader is left to hunt through the supplemental to assess what level of molecular interpretation is appropriate. It would appear that most of this information is in Extended Figure 6 where tiny little local resolution images are placed next to tiny little FSC curves. Unfortunately, the combination of small fonts, images and poor resolution means that one can barely assess the local resolution of the RQT complex and its factors. It would seem appropriate to improve this with larger images and clear local resolution for the main RQT components – average resolutions are provided in sorting schemes but these presumably reflect the resolution of the ribosome rather than the factors?
2. Extended Data Fig 9 does show some low pass filtered density for the factors, although the local resolution is not indicated. It appears that the factors range from 4-6Å, allowing only fitting of homology model rather than actually molecular modeling? Or are there regions where the register can be determined? Either way, it would be appropriate to mention this in the main text so the reader can understand the limitations of the study. Regardless, I find that the authors have been very careful to limit their interpretation to domain interactions and global motions, which would seem appropriate given the presented data.
3. I like the way the authors indicated in Fig3 using the schematics as to which parts of the Cue3 and Rqt4 were modeled, however, it was a little confusing for Slh1 – does grey indicate that only the N-terminus was not modelled...using grey for Brr2 and Ski2 confuses make this confusing.

4. Maybe I misunderstood the model, but I thought that RQT was exerting a pulling force on the mRNA and inducing back-swiveling of the head i.e. clockwise rotation when viewed from above. However, the arrows in the model have an anti-clockwise rotation?

Reviewer #2 (Remarks to the Author):

Summary

This manuscript by Best et al. studies the biochemical and structural events accompanying the action of the ribosome quality control trigger complex, RQT at ribosome stalls in budding yeast. Two novel findings are reported: 1. The main biochemical finding of the manuscript is that RQT complex requires mRNA 3' to the lead ribosome. 2. The main structural finding is that the activity of RQT complex results in an atypical swivelled head state of the lead ribosome reminiscent of a normal translocation intermediate. Using structural modeling of the RQT complex and homology to other known helicases, the authors derive a mechanism for binding and activity of the Slh1 helicase component of the RQT complex. Their primary conclusion is that the helicase activity of Slh1 acting on the mRNA results in splitting of the lead ribosome.

The biochemical and structural experiments are performed rigorously, and the above two main findings will be of interest to the translational control community. Both the requirement of ribosome collision for RQT action and the RQT-mediated dissociation of only the lead ribosome in a collision were predicted by kinetic modeling (Park Plos Biol 2019) and biochemical studies (Juszkiewicz Mol Cell 2020).

Comments

1. A major weakness of the paper is the lack of direct evidence that Slh1 helicase binds mRNA even though this is crucial to the proposed mode of RQT action. The discordance between the known structures of other similar helicases and the structural model of Slh1 action proposed here makes it essential that the authors provide direct evidence of mRNA binding by Slh1. One approach is to use UV crosslinking to show direct binding of Slh1 to the mRNA substrate in their biochemical assay. I recommend that the authors perform this experiment or use an orthogonal approach to demonstrate mRNA binding by Slh1.

2. The proposal in the abstract and the manuscript (l 28, 400, 428) that “The collided ribosome functions as a ram or giant wedge, ultimately resulting in subunit dissociation.” has no supporting evidence in this manuscript, and goes against against the authors own findings. The authors find that the pre-splitting conformation of the lead ribosome with RQT helicase-deficient mutants already has disorder in the collided Asc1 interface (Ext Fig 7b,c), arguing against RQT activity causing this disorder. In fact, if I am not mistaken, there is no evidence in the manuscript that RQT activity (as opposed to RQT binding or the preceding ubiquitylation event) on the lead ribosome requires even the presence of a collided ribosome. The atypical C2 conformation with RQTwt is observed only on an isolated 80S ribosome. The lack of a collided ribosome for the C2 conformation argues that the collided ribosome did not serve as an effective wedge. In this context, the statement in the discussion (l417) “In our study, we observe that head-swiveling directly exerts force on the trailing collided ribosome as seen in the induced disorder of the RACK1/Asc1-RACK1/Asc1 contact” is incorrect: The induced disorder is present before any head swiveling is observed and no forces are measured anywhere in the manuscript. Given these arguments, the model of collided ribosome acting as a wedge on which Slh1 helicase splits the subunits is not supported by evidence in this manuscript or in published literature, and should be modified or removed.

3. The authors’ results confirm key elements of a kinetic model first proposed in Park Plos Bio 2019 based on careful measurements of protein and mRNA level changes. Both the requirement for the collided ribosome and the preferential dissociation of the lead ribosome at collisions were predicted in this study (see collision-stimulated abortive termination model in abstract and Fig. 3 of the paper). The Park study should be cited and concordance with their conclusions should be explicitly discussed.

4. The authors use a Δ Cue2 Δ Xrn1 Δ Slh1 strain for purification, but the rationale for using the Cue2 Xrn1 deletion is not provided. This is presumably based on the results of D’Orazio 2019, and thus should be mentioned. Related point: The introduction does not include endonucleolytic cleavage as a consequence of ribosome stalling and collisions. This should be included in lines 42-44.

Reviewer #3 (Remarks to the Author):

The work by Best et al. uses cryo-EM to determine how the RQT complex releases collided ribosomes. The study is very clear and well written. It provides the first explanation on how RQT releases ribosomes, and why the complex works on the collided ribosome, but not the trailing ribosomes. I do not think the manuscript needs any revision to be published.

POINT-TO-POINT RESPONSE

Reviewer #1 (Remarks to the Author):

Ribosome collisions are recognized by the ribosome-associated quality control system and important for rescuing stalled ribosomes in all domains of life. Dissociation of the collided ribosomes is mediated by the ribosome quality control trigger complex RQT although the mechanism by which this is mediated is unknown. Here the authors provide insight into this process using a combination of in vitro biochemical splitting assays and determination of cryo-EM structures of several RQT-ribosome complexes. RQT is shown to require accessible 3' extension of the mRNA and the presence of a neighboring ribosome. Cryo-EM structures reveal that RQT engages the 40S subunit of the lead ribosome and provide the first overall architecture of RQT on the ribosome. Moreover, two conformations for RQT are observed that lead to a model whereby the RQT component Slh1 applies a pulling force on the mRNA that imparts a destabilizing conformation on the 40S subunit, with contribution of the collided ribosome that acts as a wedge to aid dissociation.

Overall, this an exciting study with providing not only a first insights into the overall architecture of RQT on the ribosome, but also mechanistic insights into how RQT mediates splitting. The manuscript is well-written and presented and the results will be of great interest to the scientific community.

I have no major concerns but only a few points for clarification.

1. In the main text, the authors skirt around the resolution of the complexes and in particular the local resolution of the factors, therefore, the reader is left to hunt through the supplemental to assess what level of molecular interpretation is appropriate. It would appear that most of this information is in Extended Figure 6 where tiny little local resolution images are placed next to tiny little FSC curves. Unfortunately, the combination of small fonts, images and poor resolution means that one can barely assess the local resolution of the RQT complex and its factors. It would seem appropriate to improve this with larger images and clear local resolution for the main RQT components – average resolutions are provided in sorting schemes but these presumably reflect the resolution of the ribosome rather than the factors?

Response:

We agree with the referee. In the original version of Supplementary Fig. 6, we aimed to fit all (local) resolution data for four datasets into one figure, resulting in many small display items. In the revised version, as requested, we focused on only the dataset, that was used for final model building. We increased the size of display items and added additional panels where the RQT components are color coded to better assess the map quality in respective regions. (Local) resolution information for all other used datasets (that were not used for modeling) was moved to respective sorting schemes (Supplementary Figs 2,3 and 5).

2. Extended Data Fig 9 does show some low pass filtered density for the factors, although the local resolution is not indicated. It appears that the factors range from 4-6A, allowing only

fitting of homology model rather than actually molecular modeling? Or are there regions where the register can be determined? Either way, it would be appropriate to mention this in the main text so the reader can understand the limitations of the study. Regardless, I find that the authors have been very careful to limit their interpretation to domain interactions and global motions, which would seem appropriate given the presented data.

Response:

As suggested by the referee, we now mention local resolution ranges for the RQT density in the main text (line 292-293). Moreover, as stated in response for the previous point, we improved Extended Data Fig. 6 (now Supplementary Figure 6), that should clarify the local resolution range for RQT components. Here, it is now clearly visible, that core regions for Slh1 are resolved down to 3.5 Å, which indeed allowed molecular modeling.

3. I like the way the authors indicated in Fig3 using the schematics as to which parts of the Cue3 and Rqt4 were modeled, however, it was a little confusing for Slh1 – does grey indicate that only the N-terminus was not modeled...using grey for Brr2 and Ski2 confuses make this confusing.

Response:

We changed the color of the Brr2 and Ski2 N-termini to light green in revised Fig. 3 and Extended Data Fig. 9, to avoid confusion. In addition, we added a sentence in the figure legend, that the Slh1 N-terminus was not modeled.

4. Maybe I misunderstood the model, but I thought that RQT was exerting a pulling force on the mRNA and inducing back-swiveling of the head i.e. clockwise rotation when viewed from above. However, the arrows in the model have an anti-clockwise rotation?

Response:

This is indeed a misunderstanding: The substrate for RQT is a POST-state ribosome (state C1). Transition from C1 to C2 results in a counterclockwise swivel motion, not a clockwise back-swiveling, as already described in the main text. Thus, the arrows describing the transition (from C1 to C2) in Fig. 4a are correct.

Reviewer #2 (Remarks to the Author):

Summary

This manuscript by Best et al. studies the biochemical and structural events accompanying the action of the ribosome quality control trigger complex, RQT at ribosome stalls in budding yeast. Two novel findings are reported: 1. The main biochemical finding of the manuscript is that RQT complex requires mRNA 3' to the lead ribosome. 2. The main structural finding is that the activity of RQT complex results in an atypical swivelled head state of the lead ribosome reminiscent of a normal translocation intermediate. Using structural modeling of the RQT complex and homology to other known helicases, the authors derive a mechanism for binding and activity of the Slh1 helicase component of the RQT complex. Their primary conclusion is that the helicase activity of Slh1 acting on the mRNA results in splitting of the lead ribosome.

The biochemical and structural experiments are performed rigorously, and the above two main findings will be of interest to the translational control community. Both the requirement of ribosome collision for RQT action and the RQT-mediated dissociation of only the lead ribosome in a collision were predicted by kinetic modeling (Park Plos Biol 2019) and biochemical studies (Juszkiewicz Mol Cell 2020).

Comments

1. A major weakness of the paper is the lack of direct evidence that Slh1 helicase binds mRNA even though this is crucial to the proposed mode of RQT action. The discordance between the known structures of other similar helicases and the structural model of Slh1 action proposed here makes it essential that the authors provide direct evidence of mRNA binding by Slh1. One approach is to use UV crosslinking to show direct binding of Slh1 to the mRNA substrate in their biochemical assay. I recommend that the authors perform this experiment or use an orthogonal approach to demonstrate mRNA binding by Slh1.

Response:

As specifically suggested by the referee, we performed UV-crosslinking experiments demonstrate mRNA binding by the RQT complex. Therefore, we reconstituted RQT with in vitro ubiquitinated disomes, stalled on a 4-thio-uridine containing mRNA and crosslinked using UV-A (specific for 4-SU). Subsequently, the crosslinked Slh1-disome complex was purified using FLAG-beads and the eluate was analyzed by Northern Blotting against the mRNA. We indeed observed that the (CGN)12 mRNA was highly enriched in the presence and upon isolation of RQT, indicating that there is a direct interaction with this mRNA when present on disomes.

We included this result in the revised Supplementary Fig. 1 as a new panel (Supplementary Fig. 1g) and rewrote the paragraph describing our in vitro binding and splitting assays accordingly (lines 117-158).

2. The proposal in the abstract and the manuscript (l 28, 400, 428) that “The collided ribosome functions as a ram or giant wedge, ultimately resulting in subunit dissociation.” has no supporting evidence in this manuscript, and goes against against the authors own findings. The authors find that the pre-splitting conformation of the lead ribosome with RQT helicase-deficient mutants already has disorder in the collided Asc1 interface (Ext Fig 7b,c), arguing against RQT activity causing this disorder. In fact, if I am not mistaken, there is no evidence in the manuscript that RQT activity (as opposed to RQT binding or the preceding ubiquitinylation event) on the lead ribosome requires even the presence of a collided ribosome. The atypical C2 conformation with RQTwt is observed only on an isolated 80S ribosome. The lack of a collided ribosome for the C2 conformation argues that the collided ribosome did not serve as an effective wedge. In this context, the statement in the discussion (l417) “In our study, we observe that head-swiveling directly exerts force on the trailing collided ribosome as seen in the induced disorder of the RACK1/Asc1-RACK1/Asc1 contact” is incorrect: The induced disorder is present before any head swiveling is observed and no forces are measured anywhere in the manuscript. Given these arguments, the model of collided ribosome acting as a wedge on which Slh1 helicase splits the subunits is not supported by evidence in this manuscript or in published literature, and should be modified or removed.

Response:

We thank the referee for his critical assessment of our model. Yet, we would like to point out that we indeed provide evidence for the requirement of a collided ribosome for RQT activity. First, we showed that in our in vitro splitting system 80S ribosomes without a neighbor are a poor substrate for RQT (Supplementary Fig. 1i in the revised manuscript). Second, while trisome and disome peaks were reduced in our in vitro splitting reactions, 80S ribosomes accumulated, indicating that they are not split by RQT (Fig. 1d in the revised manuscript). Third, in agreement with these biochemical observations, we observed in the (active) splitting reactions which were analyzed by cryo-EM large classes of RQT-bound 80S ribosomes that apparently could not be dissociated.

With respect to the destabilization of the RACK1-RACK1 interface, we were indeed logically inconsistent in the first version of the manuscript. Considering our cryo-EM data, we can only conclude that binding of the RQT complex to the disome without mRNA overhang can already destabilize the interface, yet this binding cannot be directly correlated with head-swiveling – as correctly pointed out by the referee. It is therefore likely that head swiveling and loss of RACK1 are independent destabilizing events and thus we rephrased this section in the discussion of revised manuscript (line 499 ff).

Taken together and considering the above observations, we are still convinced that it is the most plausible model that the collided second ribosome in the disome is making an important contribution to the RQT-dependent splitting reaction.

3. The authors' results confirm key elements of a kinetic model first proposed in Park Plos Bio 2019 based on careful measurements of protein and mRNA level changes. Both the requirement for the collided ribosome and the preferential dissociation of the lead ribosome at collisions were predicted in this study (see collision-stimulated abortive termination model in abstract and Fig. 3 of the paper). The Park study should be cited and concordance with their conclusions should be explicitly discussed.

Response:

We added the reference to the Park study (ref 30 in the revised manuscript) at an appropriate place of the manuscript (line 260)

4. The authors use a Δ Cue2 Δ Xrn1 Δ Slh1 strain for purification, but the rationale for using the Cue2 Xrn1 deletion is not provided. This is presumably based on the results of D'Orazio 2019, and thus should be mentioned. Related point: The introduction does not include endonucleolytic cleavage as a consequence of ribosome stalling and collisions. This should be included in lines 42-44.

Response:

As suggested, we added an explanation for the usage of the Δ Cue2 Δ Xrn1 Δ Slh1 yeast strain (line 113-115) and added the D'Orazio reference (ref 5 in the revised manuscript). Moreover, we included endonucleolytic cleavage as a consequence of ribosome stalling and collisions in the introduction (line 67-68) along with appropriate citations.

Reviewer #3 (Remarks to the Author):

The work by Best et al. uses cryo-EM to determine how the RQT complex releases collided ribosomes. The study is very clear and well written. It provides the first explanation on how RQT releases ribosomes, and why the complex works on the collided ribosome, but not the trailing ribosomes. I do not think the manuscript needs any revision to be published.

Response:

We thank the reviewer for this excellent evaluation of our work.

REVIEWERS' COMMENTS

Reviewer #2 (Remarks to the Author):

The authors have satisfactorily addressed all my comments. I congratulate the authors on a thorough study of the RQT splitting mechanism.

POINT-TO-POINT RESPONSE

REVIEWERS' COMMENTS

Reviewer #2 (Remarks to the Author):

The authors have satisfactorily addressed all my comments. I congratulate the authors on a thorough study of the RQT splitting mechanism.

Response:

We are pleased to hear that our revision of the paper was satisfactory and we thank the reviewer for this excellent evaluation of our work.